# Causally motivated multi-shortcut identification & removal

**Jiayun Zheng**
Computer Science and Engineering
University of Michigan, Ann Arbor

**Maggie Makar**[*]
Computer Science and Engineering
University of Michigan, Ann Arbor

## Abstract

For predictive models to provide reliable guidance in decision making processes, they are often required to be accurate and robust to distribution shifts. Shortcut learning–where a model relies on spurious correlations or shortcuts to predict the target label–undermines the robustness property, leading to models with poor out-of-distribution accuracy despite good in-distribution performance. Existing work on shortcut learning either assumes that the set of possible shortcuts is known *a priori* or is discoverable using interpretability methods such as saliency maps, which might not always be true. Instead, we propose a two step approach to (1) efficiently identify relevant shortcuts, and (2) leverage the identified shortcuts to build models that are robust to distribution shifts. Our approach relies on having access to a (possibly) high dimensional set of auxiliary labels at training time, some of which correspond to possible shortcuts. We show both theoretically and empirically that our approach is able to identify a sufficient set of shortcuts leading to more efficient predictors in finite samples.

## 1 Introduction

Despite their immense success, predictors constructed from deep neural networks (DNNs) tend to have poor performance under distribution shift [7, 24, 5, 14]. One reason behind such brittleness is "shortcut learning": when a predictor relies on shortcuts, i.e., spurious correlations between the inputs and the target label that are easy to learn and are predictive of the target label in the training data [15]. If these spurious correlations no longer exist when the test distribution shifts, the accuracy of the predictor deteriorates. Here, we study the problem of learning a performant predictor whose risk is invariant to interventions that change the association between shortcuts and the target label. Our work tackles two limitations in previous literature on addressing shortcut learning. First, previous work often assumes that the set of shortcuts are known in advance, or is easily identifiable using interpretability methods such as saliency maps. Second, much of the existing work assumes that there are a few (often one) shortcuts.

To tackle these limitations, we study methods to identify shortcuts, and build models that are robust (i.e., invariant) to possibly many shortcuts. Throughout, we will use the example of detecting the presence and severity of diabetic retionpathy (DR) using images taken using a funduscope. We focus on a setting where we are also given multiple auxiliary labels (e.g., the type of funduscope, patient age, sex and previous medical history) at training but not test time. A subset of these auxiliary data label factors of variation (i.e., shortcuts) that we want to be invariant to but the rest might be redundant for the purpose of shortcut removal. We propose a method to identify this subset of relevant auxiliary labels for shortcut removal, and then exploit the identified subset to construct a predictor whose risk is approximately invariant across a well-defined family of test-distributions.

---

[*]Corresponding author, email: `mmakar@umich.edu`

36th Conference on Neural Information Processing Systems (NeurIPS 2022).

Our approach can be viewed as a continuation of a line of recent work on leveraging the causal structure of a problem to build robust predictors [41, 30]. Unlike previous work, we do not assume that the relevant shortcuts are known *a priori* but instead leverage causal ideas to both identify the shortcuts and build models that are robust to these shortcuts. In addition, unlike previous work, we do not make any assumptions about the type or dimension of the auxiliary labels and the target label. Our contributions can be summarized as follows. (1) We leverage ideas from causality to show that robustness to a large set of distribution shifts is possible through ensuring invariance to a small set of shortcuts. (2) We develop a method for identifying these shortcuts, provide theoretical arguments about validity of our approach and show that it leads to more efficient predictors. (3) We extend previous work on single shortcut removal to a more general formulation that allows for high dimensional shortcuts of arbitrary types (4) We empirically validate our theoretical findings using a semi-simulated benchmark and a medical task, showing our approach has favorable in- and out-of-distribution generalization properties.

## 2   Related work

Existing work tackling out-of-distribution generalization tends to fall into two categories: those which assume access to some (usually unlabeled) examples from the target domain (e.g., [17, 20, 27, 8]) and those which do not (e.g., [39, 38, 30, 41, 36]). Our work falls into the latter category.

**Robustness to known shortcuts.** Similar to our work, a number of authors adapt causal ideas for the purpose of out-of-distribution generalization when samples from the target domain are unavailable. By contrast to our work, this line of work tends to assume that the sources of bias (or shortcuts) are known *a priori*. For example, Subbaswamy et al. [39] assume the availability of a "selection diagram" that specifies which variables have a unstable relationship with the target label, and hence could be shortcuts. Absent prior knowledge, the authors suggest constructing this selection diagrams using conditional independence tests. We show here that such tests are unreliable when the variables are high dimensional, and present an solution to this limitation. The assumption of known shortcuts is implicit in other work (e.g., [25, 36, 4, 33]) where the authors aim to find the best predictor over a set of possible distributions. Here, defining such a set requires knowledge of the meaningful shortcuts. In the experiments section, we show that our approach, by identifying a subset of relevant shortcuts, is able to outperform approaches equivalent to [36].

Unlike other work (e.g., [4, 28]), we do not assume access to data sampled from multiple environments or distributions. Instead, we assume access to auxiliary labels that may be proxies for shortcuts.

Most similar to our work is [30], where the authors study an anti-causal prediction problem similar to ours. Unlike us, they assume that there is a single shortcut labeled by a binary auxiliary label. Our work can be viewed as a direct extension of [30] to relax assumptions about the type and dimension of the auxiliary label as well as the prior knowledge about the shortcut.

**Shortcut identification.** One approach that has been suggested to identify possible shortcuts is by leveraging interpretability methods such as saliency maps [37] which visually highlight which parts of an image is most important for a prediction. However, user-based studies have found that saliency maps often have limited utility in explaining model features [2]. In addition, in domains such as healthcare, leveraging saliency maps to identify shortcuts might require expert knowledge. In [6], the authors suggest manipulating the observed examples by intervening on possible shortcuts and measuring the behviour of the model under such interventions. However, such work relies on being able to faithfully manipulate the observed data, which is not possible in most cases.

## 3   Preliminaries

**Setup.** We consider a supervised learning setup where the task is to construct a predictor $f(\mathbf{X})$ that predicts a label $Y$ (e.g., presence and severity of DR) from an input $\mathbf{X}$ (e.g., image). We assume that at training time only, we have a $d$-dimensional set of auxiliary labels $\mathbf{V}^d$. We use $V^i$ to denote the $i^{th}$ column of $\mathbf{V}^d$, and $\boldsymbol{V}^{d\setminus i}$ to denote all columns of $\mathbf{V}^d$ excluding the $i^{th}$ column. We use $\mathcal{X}, \mathcal{Y}, \mathcal{V}^d$ to denote the domains of $\mathbf{X}, Y$, and $\mathbf{V}^d$ respectively. We make no further assumptions about these domains: they can contain binary, categorical or continuous variables. We use the notation $Z \perp\!\!\!\perp_P Z'$ to denote that the two variables $Z, Z'$ are independent under the distribution $P$. Throughout, we will use capital letters to denote variables, and small letters to denote their value. Our training

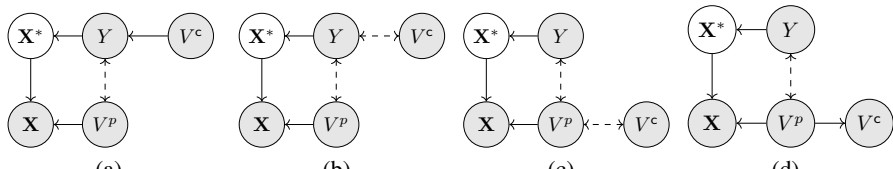

(a)       (b)       (c)       (d)

Figure 1: Examples of causal DAGs describing the setting studied in this paper. In all DAGs, the main label $Y$ and relevant auxiliary labels $\mathbf{V}^p$ generate observed input $\mathbf{X}$, redundant auxiliary labels $\mathbf{V}^c$ do not directly affect the input $\mathbf{X}$ and $Y$ only affects $\mathbf{X}$ through $\mathbf{X}^*$. In (a) $\mathbf{V}^c$ causally affects $Y$, in (b) $\mathbf{V}^c$ is correlated with $Y$ in (c) $\mathbf{V}^c$ is correlated with $\mathbf{V}^p$, and in (d) $\mathbf{V}^c$ is caused by $\mathbf{V}^p$

data consist of tuples $\mathcal{D} = \{(\mathbf{x}_i, y_i, \mathbf{v}_i^d)\}_{i=1}^n$ drawn from a source training distribution $P_s$. We will consider predictors $f$ of the form $f = h(\phi(\mathbf{x}))$, where $\phi$ is a representation mapping and $h$ is the final predictor.

We assume that $P_s$ follows an anti-causal structure, meaning that $\mathbf{X}$ is generated by the labels $Y$ and $\mathbf{V}^p$, where $\mathbf{V}^p$ is a subset of $\mathbf{V}^d$. We require that $\mathbf{V}^d$ does not contain any causal descendants of $\mathbf{X}$. We use $\mathbf{V}^c$ to denote the complement of $\mathbf{V}^p$, i.e., all the variables in $\mathbf{V}^d$ that do not directly affect $\mathbf{X}$. Importantly, we do not assume that we know *a priori* which auxiliary labels fall into $\mathbf{V}^p$ and which fall into $\mathbf{V}^c$. We assume that the labels $Y$ and $\mathbf{V}^p$ are correlated, but not causally related; that is, an intervention on $\mathbf{V}^p$ does not imply a change in the distribution of $Y$, and vice versa. Such correlation often arises through the influence of an unobserved third variable such as the environment from which the data is collected. We make no assumptions about the relationship between $Y$ and $\mathbf{V}^c$ or $\mathbf{V}^c$ and $\mathbf{V}^p$: they can be causal or correlations. Figure 1 shows examples of the causal directed acyclic graphs (DAGs) that conform with our assumptions. Solid edges in the figure depict causal relationships, and dashed bidirectional arrows depict correlations. Grey nodes are observed at training time while white nodes are unobserved.

We assume that there is a sufficient statistic $\mathbf{X}^*$ such that $Y$ only affects $\mathbf{X}$ through $\mathbf{X}^*$, and $\mathbf{X}^*$ can be fully recovered from $\mathbf{X}$ via the function $\mathbf{X}^* := e(\mathbf{X})$. However, we assume that the sufficient reduction $e(\mathbf{X})$ is unknown, so we use a white node to signify that $\mathbf{X}^*$ is unobserved in Figure 1.

In addition, we make an overlap assumption with respect to $\mathbf{V}^p$ on the source distribution, $P_s$, that is we assume that $P_s(\mathbf{V}^p)P_s(Y)$ is absolutely continuous with respect to $P_s(\mathbf{V}^p, Y)$ i.e., $P_s(\mathbf{V}^p)P_s(Y) \ll P_s(\mathbf{V}^p, Y)$. We also assume that $\mathbf{V}^p$ has a bounded variance.

To establish the intuition underlying the DAGs in figure 1, we highlight some possible scenarios that these DAGs depict. In all DAGs, $\mathbf{V}^p$ can denote the quality of the funduscope, which is used to capture the image $\mathbf{X}$, or the sex of the patient which has been shown to affect the shape of the retina [9]. In figure 1(a), $\mathbf{V}^c$ can denote high sugar intake: it can cause diabetes and its complications such as DR but it likely does not directly affect the appearance of the retina ($\mathbf{X}$) independently of $Y$. In figure 1(b), $\mathbf{V}^c$ can denote conditions that tend to co-occur with DR such as kidney diseases [31] in figure 1(c), $\mathbf{V}^c$ could be socio-economic characteristics correlated with access to high quality funduscopes (or healthcare in general) while in figure 1(d) $\mathbf{V}^c$ could be sex-specific diseases such as cervical cancer.

**Risk invariance and shortcuts.** We define the generalization risk of a function $f$ on a distribution $P$ as $R_P = \mathbb{E}_{X,Y \sim P}[\ell(f(X), Y)]$, where $\ell$ is an appropriate loss function e.g., categorical cross entropy if $Y \in \{0, \ldots, K\}$ or mean squared error if $Y \in \mathbb{R}$. We focus on obtaining an optimal *risk invariant* predictor, whose risk is invariant across a family of target distributions $\mathcal{P}$ that can be obtained from $P_s$ by interventions on the DAGs in Figure 1. Specifically, we consider interventions on any non-causal relationship that keep the marginal distribution of $Y$ constant[2]. For example, each distribution in the target family of distributions described by the DAG in figure 1(a) can be obtained by replacing the source conditional distribution $P_s(\mathbf{V}^p \mid Y)$ with a target conditional distribution $P_t(\mathbf{V}^p \mid Y)$. In this case, the target set of distributions is:

$$\mathcal{P} = \{P_s(\mathbf{X} \mid \mathbf{X}^*, \mathbf{V}^p)P_s(\mathbf{X}^* \mid Y)P_s(Y \mid \mathbf{V}^c)P(\mathbf{V}^c)P_t(\mathbf{V}^p \mid Y)\}, \tag{1}$$

This family allows the marginal dependence between $Y$ and $\mathbf{V}^p$ to change arbitrarily.

---

[2]Extending our analysis to settings where the marginal distribution of $Y$ also changes is possible, but would introduce some notational overhead. It would require that a re-weighted risk be invariant across such a family.

We define the set of risk invariant predictors to be all predictors that have the same risk for all $P_t \in \mathcal{P}$, $\mathcal{F}_{\text{rinv}} = \{f : R_{P_t}(f) = R_{P'_t}(f) \quad \forall P_t, P'_t \in \mathcal{P}\}$ and an optimal risk-invariant predictor $f_{\text{rinv}}$ to have the property $f_{\text{rinv}} \in \arg\min_{f \in \mathcal{F}_{\text{rinv}}} R_{P_t}(f) \quad \forall P_t \in \mathcal{P}$.

The definition of $\mathcal{P}$ also allows us to define a set of shortcuts that we care to remove: these are the set of shortcuts that would lead to varying risk across different distributions in $\mathcal{P}$. We will refer to this set as $\mathcal{P}$-specific shortcuts, but drop such notation when it is implied from the text.

### 3.1 The sufficiency of $\mathbf{V}^p$ for $\mathcal{P}$-shortcut removal

One of the insights of our work is that by taking into account the causal DAG that generates the data, we are able to identify a small subset of the auxiliary labels that are sufficient to induce robustness across $\mathcal{P}$. Specifically, for any DAG that satisfies the properties outlined above, we show that it is sufficient to remove shortcuts that are labeled by $\mathbf{V}^p$ to achieve robustness. We formally state this in the following proposition.

**Proposition 1.** *Let $T(P_s)$ be any transformation that renders $Y \perp\!\!\!\perp_{T(P_s)} \mathbf{V}^p$. Under such transformation, the Bayes optimal predictor is a function of $\mathbf{X}^*$ only and is asymptotically risk invariant.*

The proof of this statement follows from the fact that $\mathbf{X}^*$ d-separates $Y, \mathbf{X}$ when $Y \perp\!\!\!\perp_{T(P_s)} \mathbf{V}^p$. Since the full statement of the proof is identical that of proposition 1 in [30], it is omitted.

The proposition states that any transformation that renders $Y$ independent of $\mathbf{V}^p$ is sufficient to give us risk invariance for DAGs that satisfy the assumptions outlined above. Meaning the only shortcuts that we care about are ones induced by $\mathbf{V}^p$. Transformations $T$ include conditioning on $\mathbf{V}^p$ or reweighting the distribution. As shown in previous work, conditioning might lead to poor estimators especially when training using stochastic gradient descent with small batches [30, 29]. So we focus on reweighting schemes. We use $P^\circ$ to denote the outcome of such a reweighting transformation, i.e., $P^\circ = T(P_s)$, with $Y \perp\!\!\!\perp_{P^\circ} \mathbf{V}^p$. We refer to this $P^\circ$ as the ideal distribution. In the DR example, this distribution is one where we are equally likely to observe a man or a woman with DR.

One important consequence of proposition 1 is that it implies that overlap defined with respect to $Y, \mathbf{V}^p$ rather than $Y, \mathbf{V}^d$ is sufficient to identify robust estimators. This consequence is useful when $\mathbf{V}^d$ is high dimensional while $\mathbf{V}^p$ is low dimensional since overlap is less likely to be satisfied as the dimension of the variables increase.

## 4 Identifying a sufficient subset of shortcuts

Our training strategy follows two steps. First, we develop a novel approach to identify $\mathbf{V}^p$. Second, by extending previous work on single shortcut removal, we suggest an approach which leverages the results from the first step to train predictors that are robust to arbitrary types and dimensionality of auxiliary labels and target labels.

Our approach for identifying $\mathbf{V}^p$ leverages principles of d-separation [32]. Briefly, for an auxiliary label to be a shortcut, it must lie on an unblocked backdoor pathway between $\mathbf{X}$ and $Y$. Hence $\mathbf{V}^p$ should have an unblocked pathway to $Y$, and an unblocked pathway to $\mathbf{X}$. Our approach to identifying $\mathbf{V}^p$ relies on testing for the existence of these two pathways. We formally state this intuition in the following proposition.

**Proposition 2.** *For all $V^i \in \mathbf{V}^d$, the following two properties hold: (1) $Y \perp\!\!\!\perp_{P_s} V^i \mid \mathbf{V}^{d\setminus i} \Rightarrow V^i \notin \mathbf{V}^p$, and (2) $\mathbf{X} \not\perp\!\!\!\perp_{P_s} V^i \mid Y, \mathbf{V}^{d\setminus i} \Leftrightarrow V^i \in \mathbf{V}^p$*

Proposition 2 states that if any $V^i$ is independent of $Y$ conditional on the rest of the auxiliary variables, it is not in $\mathbf{V}^p$, and that for any $V^i$ in $\mathbf{V}^p$, it must hold that $\mathbf{X}$ is not independent of such a variable conditional on all other auxiliary labels.

These two properties provide us with two tests that enable us to identify which auxiliary labels mark shortcuts that are necessary to account for to induce robustness versus ones which are not. The first property might seem redundant since it is strictly weaker than the second property but as we show later, both properties will be helpful to efficiently identify $\mathbf{V}^p$.

In principle, we can apply nonparametric conditional independence tests to each of the auxiliary labels to identify whether it satisfies the two properties. However, the power of nonparametric

independence tests has been shown to decline as a function of of the dimension of the data [34, 35]. This dependence on the dimension of the data makes testing if $\mathbf{X} \not\perp_{P_s} V^i \mid Y, \boldsymbol{V}^{d \setminus i}$ particularly difficult in situations where $\mathbf{X}$ is high dimensional, which is the case for high resolution images.

Instead, we seek to find a low dimensional representation $s(\mathbf{X})$, with $s \in \mathcal{S}$ such that if and only if $\mathbf{X} \not\perp_{P_s} V^i \mid Y, \boldsymbol{V}^{d \setminus i}$ then it also true that $s(\mathbf{X}) \not\perp_{P_s} V^i \mid Y, \boldsymbol{V}^{d \setminus i}$. Intuitively, if $\mathbf{X}$ contains any information about a given $V^i \in \mathbf{V}^d$ in some source distribution $P_s$, $s(\mathbf{X})$ must retain such information. This intuition implies that taking $s(\mathbf{X})$ to be the empirical risk minimizing function that predicts $\mathbf{V}^d$ from $\mathbf{X}$, is a good reduction.

To prove the validity of this simple reduction, we require an assumption on the space of functions $\mathcal{S}$: we require that each variable in $\mathbf{V}^p$ is $s$-representable. Meaning there exists some $s \in \mathcal{S}$ that can perfectly predict each $V^i \in \mathbf{V}^p$. We do not require that such an $s$ is identifiable using finite samples. We note that under the causal DAGs in figure 1, for an appropriately chosen $\mathcal{S}$, there should exist performant (albeit not perfect) predictors of $\mathbf{V}^p$ from $\mathbf{X}$ since $\mathbf{V}^p$ causes $\mathbf{X}$. In cases where $\mathbf{V}^p$ is binary, the assumption of $s$-representability can be relaxed. In that case it is sufficient to assume that $\mathcal{S}$ contains some $s$ with bounded $\delta$ error such that $\delta$ is less than the proportion of the smallest subgroup defined by $Y, \mathbf{V}^p$. Under such assumption, the following proposition establishes the validity of this simple reduction.

**Proposition 3.** *For an appropriately chosen loss function $\ell$, and function space $\mathcal{S}$, let $s^*(\mathbf{X}) = argmin_{s \in \mathcal{S}} \mathbb{E}_{P_s}[\ell(s(\mathbf{X}), \mathbf{V}^d)]$. Then the following holds for all $V^i \in \mathbf{V}^d$*

$$s^*(\mathbf{X}) \not\perp_{P_s} V^i \mid Y, \boldsymbol{V}^{d \setminus i} \Leftrightarrow \mathbf{X} \not\perp_{P_s} V^i \mid Y, \boldsymbol{V}^{d \setminus i} \tag{2}$$

Proposition 2 together with proposition 3 give us a practical and efficient procedure to identify a subset of $\mathbf{V}^d$ that is sufficient for $\mathcal{P}$-shortcut removal. For each $V^i$, we propose first testing if $Y \perp_{P_s} V^i \mid \boldsymbol{V}^{d \setminus i}$. We remove labels for which this relationship holds (consistent with condition 1 of proposition 2). We use $\underline{d}$ to denote the remaining set of auxiliary label indices. For the remaining labels in $\underline{d}$, we test if the second condition of proposition 2 holds as follows. We split the training data into two sub samples $\mathcal{D}_1$ and $\mathcal{D}_2$. We use $\mathcal{D}_1$ to train a model $s : \mathbf{X} \to \boldsymbol{V}^{\underline{d}}$. We then proceed by predicting the value of $S = s(\mathbf{x}_i)$ for $i \in \mathcal{D}_2$, and testing if $S \perp V^i \mid Y, \boldsymbol{V}^{\underline{d}/ i}$ for all $i \in \underline{d}$.

To conduct the conditional independence tests, we use kernel-based conditional independence (KCIT) methods described in [43]. Such methods ascertain conditional independencies by analyzing the cross covariance operator. Intuitively, the cross-covariance operator can be thought of as an extension of the covariance matrix when the variables are infinite dimensional. We formally define it next.

**Definition 1.** *Let $Z, Z'$ be a pair of random variables defined on $\mathcal{Z} \times \mathcal{Z}'$ and let $\Omega_{\mathcal{Z}}$ and $\Omega_{\mathcal{Z}'}$ be two Reproducing Kernel Hilbert Spaces (RKHSs) defined on $\mathcal{Z}$ and $\mathcal{Z}'$. Define the cross-covariance operator of $Z, Z$, $C_{zz'} : \Omega_{\mathcal{Z}} \to \Omega_{\mathcal{Z}'}$ such that $\langle g, C_{zz'} g \rangle = Cov[g(Z), g'(Z')], \; \forall g \in \Omega_{\mathcal{Z}}, g' \in \Omega_{\mathcal{Z}'}$*

In KCIT, the cross covariance operator is used to conduct a hypothesis test with the null hypothesis defined as $s(\mathbf{X}) \perp_{P_s} V^i \mid Y, \boldsymbol{V}^{d \setminus i}$, for example in our case. We use the Gamma approximation method suggested in [43] to approximate the null distribution and reject the null if the p-value corresponding to the independence test is less than a pre-specified significance level. To account for the fact that we are conducting multiple hypothesis tests, we set the significance level to be low (0.001), following the authors of KCIT. We use the radial basis function (RBF) to estimate the kernel matrices, and use the median heuristic described in [19] to set the kernel bandwidth. Finally, KCIT requires setting a parameter $\epsilon$, which is a small regularization parameter. We set $\epsilon = 10^{-3}$ as suggested by the authors but we find that the tests are generally robust to this hyperparameter. A full description of the shortcut identification procedure is included in the appendix, section C, procedure 1.

This procedure gives us a subset of $\widehat{\mathbf{V}}^p$, which is an estimate of $\mathbf{V}^p$ that is sufficient for shortcut removal. When characteristic kernels such as the RBF are used as the basis for the RKHS over which we measure the cross covariance operator, Zhang et al. [43] show that KCIT is asymptotically consistent, which in turns mean that $\widehat{\mathbf{V}}^p$ is an asymptotically consistent estimate of $\mathbf{V}^p$.

# 5 Building risk invariant predictors

Given the identified set $\widehat{\mathbf{V}}^p$, the challenge of building an invariant predictor reduces to an extension of Makar et al. [30]. In that work, the authors study a more restrictive setting where it is assumed that $\mathbf{V}^c = \emptyset$. They develop a reweighting scheme and a causally-motivated regularization scheme that lead to efficient and asymptotically robust predictors. However, their reweighting scheme assumes that the auxiliary and target labels are binary, while the regularization scheme assumes that there is a single, binary auxiliary label. We extend both components of the training procedure to a more general setting with no restrictions on the dimension or type of auxiliary and target labels.

**Reweighting to recover $P^\circ$.** Guided by our findings from proposition 1, and similar findings in [30], we reweight data sampled from an arbitrary $P_s$ to generate a pseudo-sample from $P^\circ$. As proposition 1 states, the Bayes optimal predictor under this reweighted distribution is robust to the shortcuts. Unfortunately, the reweighting scheme suggested by Makar et al. [30] does not extend to our setting, where $\widehat{\mathbf{V}}^p$ can be an arbitrary (rather than binary) high dimensional (rather than single dimensional) variable. Instead of defining the sample weights to be $u^{\mathsf{bin}}(y_i, \widehat{\mathbf{v}}_i^p) = \frac{P_s(Y=y_i)P_s(\widehat{\mathbf{V}}^p=\widehat{\mathbf{v}}_i^p)}{P_s(Y=y_i,\widehat{\mathbf{V}}^p=\widehat{\mathbf{v}}_i^p)}$, which assumes that $\widehat{\mathbf{V}}^p$ and $Y$ are binary, we leverage permutation weighting [3] which allows for arbitrarily valued $\widehat{\mathbf{V}}^p$ and $Y$. Permutation weighting proceeds by permuting $Y$ in the training data to create $\mathcal{D}' = \{(\mathbf{x}_i, y_{\pi(i)}, \mathbf{v}_i^d)\}_{i=1}^n$, where $\pi$ is a random permutation of the indices. Such a permutation mimics the desirable independencies in $P^\circ$ by breaking any correlations between $Y$, and $\widehat{\mathbf{V}}^p$. The original $\mathcal{D}$ and the permuted $\mathcal{D}'$ are stacked and a label $C \in \{0, 1\}$ is given to examples in the observed and permuted data respectively. A classifier $\eta : \mathcal{Y} \times \mathcal{V}^p \to \{0, 1\}$ is trained to learn $P_s(C = 1 \mid Y, \widehat{\mathbf{V}}^p)$. The final weights are then computed as:

$$u_i = \frac{\eta(\widehat{\mathbf{v}}_i^p, y_i)}{1 - \eta(\widehat{\mathbf{v}}_i^p, y_i)} = \frac{P_s(C = 1 \mid \widehat{\mathbf{v}}_i^p, y_i)}{P_s(C = 0 \mid \widehat{\mathbf{v}}_i^p, y_i)}. \tag{3}$$

We use $\tilde{u}_i$ to denote a normalized version of $u_i$ such that $\sum_i \tilde{u}_i = 1$. As Arbour et al. [3] show, $u_i = \frac{P_s(y_i)P_s(\widehat{\mathbf{v}}_i^p)}{P_s(y_i,\widehat{\mathbf{v}}_i^p)} = \frac{dP^\circ}{dP_s}$. Hence, under this reweighting scheme, the empirical risk minimizer $f^* = \operatorname{argmin}_f \sum_i \tilde{u}_i \ell(f(\mathbf{x}_i), y_i)$ is asymptotically risk invariant. The proof for this statement is identical to results by Makar et al. [30] and is therefore omitted.

**Causally-motivated regularization for lower variance.** While reweighting gives asymptotically robust estimators, such estimators tend to have higher variance, i.e., they are inefficient in finite samples [10]. Following Makar et al. [30], we propose a regularization scheme that leads to more efficient predictors by leveraging findings from proposition 1. This proposition establishes that under $P^\circ$, the optimal risk invariant predictor is a function of $\mathbf{X}^*$ only and hence encodes the following independence property: $\phi(\mathbf{X}) \perp\!\!\!\perp \widehat{\mathbf{V}}^p$. As a result, we consider penalizing models which do not encode this independence property. To do so, we will leverage the Hilbert Schmidt Independence Criterion (HSIC). For two arbitrary variables $Z, Z'$, the HSIC is defined as the squared Hilbert-Schmidt (HS) norm of their cross covariance operator $C_{zz'}$, defined in definition 1, i.e. $\mathrm{HSIC}(Z, Z') := \|C_{zz'}\|_{\mathrm{HS}}^2$. The HSIC measures the magnitude of the correlation between infinite dimensional projections of two arbitrary variables $Z, Z'$. As before, we use the RBF kernel when estimating the HSIC.

For data sampled from $P^\circ$, we can use the HSIC to enforce $\phi(\mathbf{X}) \perp\!\!\!\perp \widehat{\mathbf{V}}^p$ by penalizing $\mathrm{HSIC}(\phi(\mathbf{X}), \widehat{\mathbf{V}}^p)$. However, in the more likely case where $P_s \neq P^\circ$, we need to penalize a weighted version of the HSIC. This weighting in necessary since the independence property only holds under $P^\circ$. Specifically, we use the weighted HSIC estimator suggested by Hu et al. [23] (see their Proposition 3) [3].

Putting all components of our approach together the final objective to optimize is

$$h^*, \phi^* = \operatorname*{argmin}_{h,\phi} \sum_i \tilde{u}_i \ell(h(\phi(\mathbf{x}_i)), y_i) + \alpha \cdot \widehat{\mathrm{HSIC}}_\gamma^{\boldsymbol{u}}(\phi(\mathbf{X}), \widehat{\mathbf{V}}^p), \tag{4}$$

---

[3]The HSIC estimator we use here has a finite sample bias of $O(n^{-1})$, which is negligible in light of the finite sample fluctuations that dominate the convergence rate. We use this biased estimator because it is more efficient to estimate and is more commonly used in the literature.

where $\alpha > 0$ is a hyperparameter that controls the cost of violating the HSIC penalty, $\widehat{\text{HSIC}}_\gamma^{\boldsymbol{u}}$ is the estimate of the HSIC, computed over samples weighted by $\tilde{u}$ which is defined in equation (3) using a kernel with bandwidth $\gamma$. In contrast to Makar et al. [30], by regualrizing the HSIC rather than the Maximum Mean Discrepancy, our approach allows for arbitrary types of auxiliary labels with large dimensions. In the appendix, we show that this improvement does not come at the cost of statistical efficiency by showing that our estimator inherits the finite sample efficiency guarantees of the methods described in [30].

**Cross-validation.** The objective function in (4) depends on two hyperparameters: the cost of the HSIC penalty $\alpha$, and the penalty's kernel bandwidth $\gamma$. Unlike many regularizers, the HSIC penalty depends on the distribution of the data, and is vulnerable to overfitting, such that the estimated $\widehat{\text{HSIC}}$ on the training data underestimates the population HSIC. For this reason, we follow a two-step cross-validation procedure. Letting $\mathcal{D}_{\text{valid}}$ denote a held out validation set, $\phi_{\text{valid}}$ denote $\{\phi(\mathbf{x}_i)\}_{i \in \mathcal{D}_{\text{valid}}}$, and similarly define $\widehat{\mathbf{V}}_{\text{valid}}^p$, our cross validation procedure proceeds as follows. In the first step, for a given $\alpha = \alpha_0, \gamma = \gamma_0$, we first check if the corresponding $\phi_{\text{valid}}$ is independent of $\widehat{\mathbf{V}}_{\text{valid}}^p$. We do so using the permutation test suggested by Gretton et al. [19]. This test entails creating 100 permutations of the validation set, with the $k^{th}$ permutation defined as $\mathcal{D}' = \{\mathbf{x}_i, y_i, \widehat{\mathbf{v}}_{\pi^k(i)}^p\}$, and $\pi^k(i)$ is a permutation of the indices. We compute a vector of HSIC values for each of the permuted datasets, and the corresponding $1 - \beta^{\text{th}}$ quantile of that vector. $\beta$ is a pre-specified significance level that we use to accept or reject the null hypothesis that the estimated $\phi(\mathbf{X}), \mathbf{V}^p$ are independent. Similar to before, we set that to be 0.001 as a heuristic to account for the multiple tests. We reject $\alpha_0, \gamma_0$ as valid hyperparameters if $\widehat{\text{HSIC}}$ as calculated on the unpermutated validation set is larger than the value corresponding to the $1 - \beta^{\text{th}}$ quantile. Repeating this process for all $\alpha, \gamma$ candidates gives us a subset of the hyperparameters that lead to models encouraging the desired invariances. In the second step, we pick the best performing model out of this subset of candidate functions.

Pseudocode for our full approach is included in the appendix section C.

**Remarks:**

**1. Invariance to the full $\mathbf{V}^d$.** We note that it is possible to bypass the first step of our approach–identifying $\mathbf{V}^p$–and define the weights in equation 3, and the HSIC with respect to $\mathbf{V}^d$. Such a predictor might still be asymptotically robust but it will have higher variance than an estimator that relies on $\mathbf{V}^p$ only for two reasons: first, when $d > p$, $u$ as defined with respect to $\mathbf{V}^d$ will be less stable due to conditioning on a larger set of variables. In the appendix, we discuss how this might translate into a less favorable generalization error bound. Second, the power of the HSIC estimation problem decline as a function of of the dimension of the data [34, 35], making our regularizer less reliable in small samples. We empirically validate the limitations of bypassing the first step of our approach in section 6.

**2. Errors in $\widehat{\mathbf{V}}^p$.** While it is true that the two independence tests outlined in section 4 are asymptotically consistent, meaning $\widehat{\mathbf{V}}^p$ should converge to $\mathbf{V}^p$ as the sample size goes to $\infty$, it is possible that $\widehat{\mathbf{V}}^p \neq \mathbf{V}^p$ due to finite sample variability. Under some additional assumptions, it can be shown that the generalization error bound of our proposed estimator has a fourth order (i.e., mild) dependence on errors in $\widehat{\mathbf{V}}^p$, following results by Foster and Syrgkanis [13]. The details of this analysis are left as future work.

# 6 Experiments

We empirically test the two main claims in this paper: (1) that our approach is able to identify a sufficient set of auxiliary labels to induce robustness to shortcuts, and (2) that our approach leads to invariant predictors in settings where the target label and/or the auxiliary labels are high dimensional and/or non-binary. We study two different tasks: predicting bird types from images, and predicting diabetic retinopathy from fundus images. Throughout, we will evaluate the performance of our approach vis-a-vis baseline methods by comparing the area under the receiver operating curve (AUROC) on a set of shifted test distributions sampled from the family described in equation 1. Our code is available on `https://github.com/mymakar/cm_multishortcut_id_removal`.

## 6.1 Waterbirds

**Setup.** The goal of this setting is to test if our approach is able to identify shortcuts and in turn lead to more efficient predictors. Our data generation process follows the DAG described in figure 1(a), where we have a high dimensional set of auxiliary labels with a small subset that affects both the outcome $Y$ and the image $\mathbf{X}$ while the rest only affect $Y$. We follow Sagawa et al. [36] by constructing a semi-synthetic waterbirds dataset where the task is to predict $Y$, the type of bird (land or water). In this setting $\mathbf{V}^p$ is 2 dimensional, with $V^{p0}$ representing the image background (land or water) and $V^{p1}$ camera artifacts (present or absent). To generate the background shortcut, we combine images of water and land birds extracted from the Caltech-UCSD Birds-200-2011 (CUB) dataset [42] with water and land background extracted from the Places dataset [44]. To generate the camera artifact shortcut, we add small black patches to the image if camera artifacts are present. In addition, we generate 10 auxiliary labels ($\mathbf{V}^c$) that affect the outcome $Y$ but not the image $\mathbf{X}$. All labels in this example ($Y$, $\mathbf{V}^p$ and $\mathbf{V}^c$) are binary. Additional details about the data generation process and examples of the generated images are included in the appendix.

We generate the source distribution $P_s$ such that $P_s(V^{p0} = 1 \mid Y = 1) = P_s(V^{p0} = 0 \mid Y = 0) \approx 0.75$, and $P_s(V^{p1} = 1 \mid Y = 1) = P_s(V^{p1} = 0 \mid Y = 0) \approx 0.65$. We also generate three test distributions: $P_s$, $P_{\text{Flip}}$, and $P^\circ$. $P_s$ is the same as the training distribution. It serves to show us how models perform in-distribution. $P^\circ$ is the ideal distribution, where $P^\circ(V^{p0} = 1 \mid Y = 1) = P^\circ(V^{p0} = 0 \mid Y = 0) = P^\circ(V^{p1} = 1 \mid Y = 1) = P^\circ(V^{p1} = 0 \mid Y = 0) = 0.5$. It presents a test on how models perform with some deviation from the training distribution. Finally, $P_{\text{Flip}}$ is the most dissimilar to the training distribution, where the relationship between $V^{p0}, V^{p1}$ and $Y$ is flipped in that $P_{\text{Flip}}(V^{p0} = 1 \mid Y = 1) = P_{\text{Flip}}(V^{p0} = 0 \mid Y = 0) \approx 0.25$, and $P_{\text{Flip}}(V^{p1} = 1 \mid Y = 1) = P_s(V^{p1} = 0 \mid Y = 0) \approx 0.35$. The relationship between $\mathbf{V}^c$ and $Y$ is the same across all three test distributions. We introduce noise by randomly flipping 1% of the labels.

We use ResNet-50 [22], pretrained on ImageNet [12]. All models in this paper are implemented in TensorFlow [1]. We present the results from 10 simulations. In each simulation, we generate different train/test splits, different draws of auxiliary labels and different bird-background-camera artifact combinations. Additional details about training are included in the appendix.

**Baselines.** We compare our approach to the following baselines: (1) **L2** is the standard neural network trained to minimize the empirical risk, with an $L2$ penalty on the model weights. (2) **W-L2-FullV** minimizes the weighted empirical risk, with the weights computed as defined in equation 3 but using the full set of 12 auxiliary variables, $\mathbf{V}^d$. (3) **W-L2-S** is similar to W-L2-FullV but it follows the first step in our approach to first identify a sufficient set of auxiliary labels to compute the sample weights. (4) **W-L2-HDX** is similar to W-L2-S but it does not leverage our findings in proposition 3, i.e., instead of first reducing $\mathbf{X}$ to the low dimensional $s(\mathbf{X})$, it conducts the conditional independence tests on the raw input $\mathbf{X}$. (5) **W-HSIC-FullV** is similar to W-L2-FullV but instead of an L2 penalty, it penalizes the HSIC penalty defined with respect to the full $\mathbf{V}^d$ (6) **W-HSIC-HDX** is similar to W-L2-HDX but it penalizes the HSIC penalty defined with respect to the set of auxiliary labels identified based on conditional tests on the raw input $\mathbf{X}$, without using our $s(\mathbf{X})$ reduction.

Note that as Sagawa et al. [36] show, the baselines W-L2-FullV, W-L2-S and W-L2-HDX are equivalent to distributionally robust optimization in some special cases.

**Results.** We find that by reducing $\mathbf{X}$ to its low dimensional sufficient statistic, our approach is able to correctly identify the two true auxiliary labels which mark the true shortcuts in all 10 simulations. By contrast, utilizing the full $\mathbf{X}$ rather than $s(\mathbf{X})$ to conduct the conditional independence tests identifies the correct auxiliary labels in 1 out of the 10 simulations, and for the remaining 9 it is able to identify only one of the two auxiliary labels.

Figure 2 shows the predictive performance of each of the models as measured by the AUROC ($y$-axis), on the three different test distributions $P_{\text{Flip}}$, $P^\circ$, and $P_s$ ($x$-axis). We find that our approach outperforms all others under distribution shift and performs comparably to the best models in-distribution. As expected, the L2 model performs well only in-distribution but its performance quickly deteriorates out of distribution signaling a reliance on the shortcuts. All models penalizing the HSIC penalty perform better than their L2 regularized counterparts signaling that the HSIC penalty is successful in leading to more efficient estimators. W-HSIC-HDX and W-HSIC-FullV are unable to achieve the same level of robustness as our approach highlighting the limitation of conducting the conditional independence tests on the full, "unreduced" $\mathbf{X}$, and the importance of selecting a

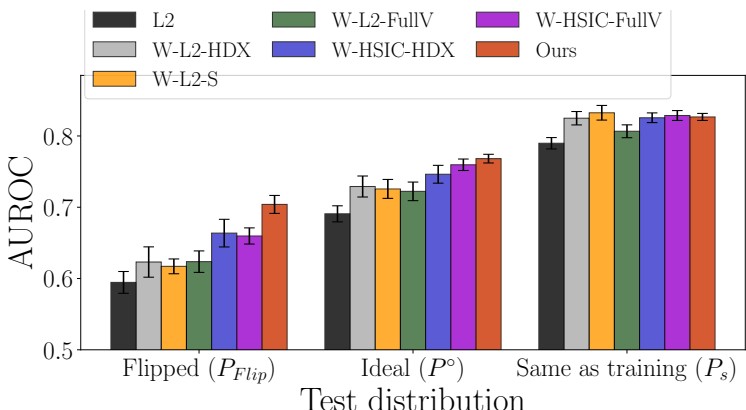

Figure 2: Waterbirds results. $x-$axis shows the test distribution, $y-$axis shows the AUROC. Our approach outperforms others in the most severe distribution shifts (flipped distribution) and performs comparably to others in-distribution.

sufficient subset of shortcuts respectively. However, these two models still perform better than models which do not include the HSIC penalty, signifying some robustness to incorrect estimates of $\widehat{\mathbf{V}}^p$.

While in principle the L2 model should outperform others when the test distribution is the same as the training distribution, it–somewhat surprisingly–does not. This could be explained by the fact that the HSIC regularized models are more efficient in finite samples, as suggested by proposition A1 in the appendix, section B. However, better performance of the L2 model in distribution would likely translate into even worse performance in shifted distributions such as the "flipped" distribution.

## 6.2 Diabetic Retinopathy

**Setup.** In this setting, we examine the validity of our approach when the outcome is non-binary. We use a publicly available dataset made available by EyePACS, LLC [11][4]. Here, we predict the presence and severity of diabetic retinopathy (DR) using fundus images, with $Y \in \{0, \ldots, 4\}$. To focus the analysis on the challenges pertaining to categorical outcomes, we generate a single binary auxiliary label, $V^p$, reflecting the presence or absence of funduscope artifacts. Similar to before, we add small black patches to the image if funduscope artifacts are present. We simulate the training distribution $P_s$ with $P_s(V^p = 1 \mid Y = 0) = P_s(V^p = 0 \mid Y > 0) = 0.9$. We introduce noise by randomly permuting 1% of the labels.

Here, we compare two baselines to our approach: L2 is defined similar to before, W-L2 is a weighted version of L2, using weights defined with respect to $V^p$. We follow Li et al. [26] in using an Inception-V3 architecture [40] to train all models. We present the results from 10 simulations. In each simulation, we generate different train/test splits and different draws of auxiliary labels.

Similar to the waterbirds setting, we measure the performance of the three models on three distributions $P_s$, $P_{\text{Flip}}$, and $P^\circ$, where $P_{\text{Flip}}$ has $P_{\text{Flip}}(V^p = 1 \mid Y = 0) = P_{\text{Flip}}(V^p = 0 \mid Y > 0) = 0.1$ and $P^\circ$ is the ideal distribution.

**Results.** Table 1 shows the AUROCs averaged over 10 simulations and their corresponding standard errors. The results show that our approach vastly outperforms others in the most severe distribution shifts, and performs relatively on par with the other models in-distribution. The slight drop in accuracy in-distribution is attributable to the fact that the baselines exploit the shortcut whereas our approach does not. The results confirm that our approach extends to setting where the target label is non-binary.

---

[4]Approval for the use of this data set for the purpose of research was obtained via correspondence with the data curators.

| Model | AUROC (STE) | | |
| --- | --- | --- | --- |
| | Flipped ($P_{\text{Flip}}$) | Ideal ($P^\circ$) | Same ($P_s$) |
| L2 | 0.69 (0.009) | 0.82 (0.003) | **0.92 (0.001)** |
| W-L2 | 0.68 (0.015) | 0.82 (0.005) | **0.92 (0.001)** |
| Ours | **0.72 (0.026)** | **0.83 (0.007)** | 0.91 (0.007) |

Table 1: Diabetic retinopathy results: AUROCs averaged over 10 simulations and standard deviations across 3 test distributions. Our approach outperforms others especially when the distribution shift is most severe, and performs comparably to others in-distribution

## 7 Conclusion

We presented an approach to identify a sufficient set of shortcuts and leverage the identified shortcuts to build predictors that are invariant to distribution shifts. Guided by insights from the causal DAG underlying the prediction problem, we analyzed the theoretical properties of our suggested approach, showing that it is both consistent and efficient. Empirically, we showed that our approach outperforms others using a semi-simulated dataset and a medical dataset.

**Limitations.** One of the strengths of our approach is identifying a small subset of relevant shortcuts. In doing that, we were able to weaken the overlap assumption relative to an approach that treats all auxiliary labels as possible shortcuts. However, in cases where $\mathbf{V}^p$ is high dimensional, this weaker overlap assumption might be violated, especially with small samples. One way to address this limitation is by first checking if overlap is satisfied. Absent strong assumptions or additional data, our approach (and any other learning-based approach) will not be able to generalize to subgroups for which overlap is violated.

**Societal impact.** Our approach could be used in fairness applications where invariance to auxiliary sensitive labels is desired. We caution that like any machine learning-based predictor, our approach is imperfect in that it might still encode some biases. In addition, when used for the purpose of fairness, the first step of our approach might not be desirable: practitioners might wish to enforce invariance with respect to a pre-specified sensitive label rather than a learned label.

## Acknowledgments and Disclosure of Funding

We would like to thank Alex D'Amour, Michael Dykstra and the anonymous reviewers for their comments and feedback. This work was funded by the National Science Foundation under Grant No. 2153083. Any opinions, findings, and conclusions or recommendations expressed in this material are those of the author(s) and do not necessarily reflect the views of the National Science Foundation.

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
