# A Proofs for section 4

## A.1 Proof for proposition2

The proof relies on examining the d-separation properties implied by the DAGs. We will assume that $V^p$ and $V^c$ are single dimensional for simplicity.

**First property.** Note that for all DAGs, the path $Y \leftarrow\!-\!-\!\rightarrow V^i$ is unblocked by any other $\boldsymbol{V}^{d\setminus i}$, which means that for all $V^i \in \mathbf{V}^p$, $Y \not\perp\!\!\!\perp V^i \mid \boldsymbol{V}^{d\setminus i}$.

**Second property.** We start by proving the direction: $\mathbf{X} \not\perp\!\!\!\perp_{P_s} V^i \mid Y, \boldsymbol{V}^{d/\ i} \Rightarrow V^i \in \mathbf{V}^p$. Suppose that there exists some $V^i$ such that $\mathbf{X} \perp\!\!\!\perp_{P_s} V^i \mid Y, \boldsymbol{V}^{d/\ i}$ but $V^i \in \mathbf{V}^p$. This means that all paths between $V^i$ and $\mathbf{X}$ are blocked by $\boldsymbol{V}^{d/\ i}, Y$. However by assumption that $\mathbf{V}^p$ is a causal parent of $\mathbf{X}$, the the path $\mathbf{X} \leftarrow V^p$ cannot be blocked via conditioning on any other variables, which represents a contradiction.

We next prove the direction $V^i \in \mathbf{V}^p \Rightarrow \mathbf{X} \not\perp\!\!\!\perp_{P_s} V^i \mid Y, \boldsymbol{V}^{d/\ i}$. Suppose there exists some $i$ such that $V^i \notin \mathbf{V}^p$ but $\mathbf{X} \not\perp\!\!\!\perp_{P_s} V^i \mid Y, \boldsymbol{V}^{d/\ i}$. Then there exists a path between $V^i$ and $\mathbf{X}$ that is unblocked by $Y, \boldsymbol{V}^{d/\ i}$. In this case, $V^i$ must be a parent of $\mathbf{X}$, which is a contradiction of $V^i \notin \mathbf{V}^p$, or a child of $\mathbf{X}$ which is a contradiction of the assumptions outlined in section 3.

## A.2 Proof for proposition 3

The direction $\mathbf{X} \not\perp\!\!\!\perp_{P_s} V^i \mid Y, \boldsymbol{V}^{d/\ i} \Rightarrow s^*(\mathbf{X}) \not\perp\!\!\!\perp_{P_s} V^i \mid Y, \boldsymbol{V}^{d/\ i}$ is easy to prove as follows. Suppose that $\mathbf{X} \perp\!\!\!\perp_{P_s} V^i \mid Y, \boldsymbol{V}^{d/\ i}$ but $s^*(\mathbf{X}) \perp\!\!\!\perp_{P_s} V^i \mid Y, \boldsymbol{V}^{d/\ i}$. This statement presents an immediate contradiction since any functions of independent random variables must be independent so such an $s^*$ cannot exist.

Next, the direction $s^*(\mathbf{X}) \not\perp\!\!\!\perp_{P_s} V^i \mid Y, \boldsymbol{V}^{d/\ i} \Rightarrow \mathbf{X} \not\perp\!\!\!\perp_{P_s} V^i \mid Y, \boldsymbol{V}^{d/\ i}$. Suppose that $s^*(\mathbf{X}) \perp\!\!\!\perp_{P_s} V^i \mid Y, \boldsymbol{V}^{d/\ i}$ but $\mathbf{X} \not\perp\!\!\!\perp_{P_s} V^i \mid Y, \boldsymbol{V}^{d/\ i}$. By proposition 2, $\mathbf{X} \not\perp\!\!\!\perp_{P_s} V^i \mid Y, \boldsymbol{V}^{d/\ i}$ implies that $V^i \in \mathbf{V}^p$. And by the assumption that $\mathbf{V}^p$ is $s$-representable, we have that there exists some $s$ such that $V^i = s(\mathbf{X})$. Such an $s$ is an empirical risk minimizer achieving the minimum possible risk of 0. By definition for such an $s$, $s(\mathbf{X}) \not\perp\!\!\!\perp_{P_s} V^i \mid Y, \boldsymbol{V}^{d/\ i}$ and $\mathbb{E}_{P_s}[s(\mathbf{X})|V^i = v^i, Y = y, \boldsymbol{V}^{d/\ i} = \boldsymbol{v}^{d/\ i}] \neq \mathbb{E}_{P_s}[s(\mathbf{X})|V^i = \tilde{v}^i, Y = y, \boldsymbol{V}^{d/\ i} = \boldsymbol{v}^{d/\ i}]$ for all $v^i \neq \tilde{v}^i$ and all $z, y$. However, for $s^*(\mathbf{X}) \perp\!\!\!\perp_{P_s} V^i \mid Y, \boldsymbol{V}^{d/\ i}$ to hold, it must be true that $\mathbb{E}_{P_s}[s(\mathbf{X})|V^i = v^i, Y = y, \boldsymbol{V}^{d/\ i} = \boldsymbol{v}^{d/\ i}] = \mathbb{E}_{P_s}[s(\mathbf{X})|V^i = \tilde{v}^i, Y = y, \boldsymbol{V}^{d/\ i} = \boldsymbol{v}^{d/\ i}]$ for all $v^i \neq \tilde{v}^i$ and all $z, y$. This means that $s^*$ must have an empirical risk greater than 0, i.e., it is not an empirical risk minimizer which is a contradiction.

# B Proofs for section 5

## B.1 Reducing sample complexity

To explain how the HSIC penalty leads to a reduction in the sample complexity and hence the variance of the estimator, we follow the same strategy as Makar et al. [30] in studying a simple setting where we focus on a linear function class and analyze how the suggested HSIC penalty compares to a standard $L2$-regularized function class. Our analysis is extendable to more complex neural networks e.g., through approaches studied in [16].

For some $A > 0, \tau \geq 0$, define the two function classes:

$$\mathcal{F}_{L_2} := \{f : \mathbf{x} \mapsto \sigma(\mathbf{w}^\top \mathbf{x}), \|\mathbf{w}\|_2 \leq A\}, \tag{5}$$

$$\mathcal{F}_{L_2,\text{HSIC}} := \{f : \mathbf{x} \mapsto \sigma(\mathbf{w}^\top \mathbf{x}), \|\mathbf{w}\|_2 \leq A, \text{HSIC} \leq \tau\}. \tag{6}$$

In this simple function class, the HSIC constraint restricts the projection of the weights $\mathbf{w}$ onto $\Delta := \text{Cov}_{P\circ}(\mathbf{X}, V^p)$. To simplify notation, we assume that the variance of $\mathbf{V}^p$ is the vector of ones. However, if that is not true, $\mathbf{V}^p$ can be rescaled such that the variance is the vector of ones, which is possible because of the assumption of bounded variance. The matrix $\Delta$ is the average change in

$\mathbf{X}$ caused by intervening to change each dimension in $\mathbf{V}^p$ under $P^\circ$. Define the projection matrix $\Pi := \Delta(\Delta^\top \Delta)^{-1}\Delta^\top = \|\Delta\|_2^{-2}\Delta\Delta^\top$, which projects any vector onto $\Delta$, and $\mathbf{w}_\perp := \Pi\mathbf{w}$ as the projection of $\mathbf{w}$ onto $\Delta$, which can be thought of as the "irrelevant" dimension of $\mathbf{X}$. To directly compare our results with Makar et al. [30], we consider the case where $\mathbf{V}^p$ is one dimensional, i.e., $\Delta$ is a vector.

**Proposition A1.** *Let* $f(\mathbf{x}) = \sigma(\phi(\mathbf{x})) = \sigma(\mathbf{w}^\top \mathbf{x})$ *be a function contained in* $\mathcal{F}_{L_2,\text{HSIC}}$*. Then,* $\|\mathbf{w}_\perp\| \leq \frac{\tau}{\|\Delta\|}$*. and*

$$\mathfrak{R}(\mathcal{F}_{L_2}) \leq \frac{A\sqrt{B_\|^2 + B_\perp^2}}{\sqrt{n}}, \quad and \quad \mathfrak{R}(\mathcal{F}_{L_2,\text{HSIC}}) \leq \frac{A \cdot B_\| + \tau\frac{B_\perp}{\|\Delta\|}}{\sqrt{n}}.$$

*Proof.* By Gretton and Györfi [18] we have that:

$$\text{HSIC}(\phi(\mathbf{X}), \mathbf{V}^p, \Omega, \Psi) \geq \sup_{\omega \in \Omega, \psi \in \Psi} \|\text{Cov}(\omega(\phi(\mathbf{X})), \psi(\mathbf{V}^p))\|_{HS},$$

where $\Omega$ and $\Psi$ are two RKHS spaces defined over $\phi(\mathbf{X})$ and $\mathbf{V}^p$ respectively.

Taking $\omega$, and $\psi$ to be the identity functions, and substituting $\phi$ for $\mathbf{w}^\top \mathbf{x}$, we have that:

$$\begin{aligned}
\tau &\geq \text{HSIC}(\phi(\mathbf{X}), \mathbf{V}^p, \Omega, \Psi) \\
&\geq \|\text{Cov}(\mathbf{w}^\top \mathbf{X}, \mathbf{V}^p)\|_F \\
&= \|\mathbf{w}\text{Cov}(\mathbf{X}, \mathbf{V}^p)\|_2 \\
&= \|\mathbf{w}\Delta\|_2 \\
&= |\mathbf{w}\Delta|
\end{aligned}$$

where $\|.\|_F$ is the Frobenius norm, and the equalities follow from the fact that $\mathbf{w}\text{Cov}(\mathbf{X}, \mathbf{V}^p)$ is a scalar. Note that $\|\mathbf{w}_\perp\| = \frac{|\mathbf{w}\Delta|}{\|\Delta\|}$, which completes our proof for the first part of the statemenet (bound on $\|\mathbf{w}_\perp\|$). The rest of the proof follows identically to Makar et al. [30]. $\qquad\square$

The generalization bound can also be obtained identically to Makar et al. [30], and is hence omitted. We note that if $\mathbf{V}^d$ is used instead of $\mathbf{V}^p$, the term $C_P$, which represents a bound on $\frac{P^\circ(Y|\mathbf{V}^d)}{P_s(Y|\mathbf{V}^d)}$, in proposition A8 of [30] is likely larger, leading to a larger (less favorable) generalization error bound.

## C  Summary of proposed algorithm

Here we present pseudo code for our main approach in Algorithm 1, which calls procedure 1 as a sub-routine. Procedure 1 calls on KCIT as a subroutine. KCIT is the conditional kernel independence test described in detail in Zhang et al. [43]. It takes as an input a tuple of variables, $A, B, C$ and tests if $A \perp\!\!\!\perp B \mid C$. KCIT requires specifying a bandwidth for the kernels used. We follow the median heuristic as the authors suggest. In addition, Procedure 1 calls Cross-validation as a subroutine. Cross-validation is any generic subroutine such as $K-$fold cross validation designed to identify the most accurate model out of a set of candidate models.

**Algorithm 1:** Pseudocode for our approach

---

**Input** : $\mathcal{D}, \boldsymbol{\lambda}, \boldsymbol{\alpha}, \boldsymbol{\gamma}$
**Output** : $f^*$
```
/* Identify a sufficient set of auxiliary labels using Procedure 1 */
```
1   $\widehat{V} = \text{IdentifySufficientSet}(\mathcal{D}, \boldsymbol{\lambda})$
2   Split $\mathcal{D}$ into $\mathcal{D}_{\text{u}}$ and $\mathcal{D}_{\text{learn}}$
```
/* Compute weights, following [3] */
```
3   $\mathcal{D}_{\text{perm}} := \{\mathbf{x}_i, y_{\pi(i)}, \widehat{\boldsymbol{v}}_i\}_{i \in \mathcal{D}_{\text{u}}}$
4   Assign label $C = 1$ for $i \in \mathcal{D}_{\text{perm}}$ and $C = 0$ for $i \in \mathcal{D}_{\text{u}}$
5   Train a classifier: $g : Y, \widehat{V} \rightarrow C$
6   Compute $\boldsymbol{u}_{\text{learn}} = \{u_i\}_{i \in \mathcal{D}_{\text{learn}}} = \{g(y_i, \widehat{\boldsymbol{v}}_i)\}_{i \in \mathcal{D}_{\text{learn}}}$
```
/* Cross-validation and training */
```
7   Split $\mathcal{D}_{\text{learn}}$ into $\mathcal{D}_{\text{tr}}$ and $\mathcal{D}_{\text{valid}}$
8   Create corresponding splits for the weights: $\boldsymbol{u}_{\text{learn}}$ split into $\boldsymbol{u}_{\text{tr}}$ and $\boldsymbol{u}_{\text{valid}}$
9   Set ValidHyperparams $= \emptyset$
10 **foreach** $\alpha, \gamma \in \boldsymbol{\alpha}, \boldsymbol{\gamma}$ **do**
11     Using $\mathcal{D}_{\text{tr}}, \boldsymbol{u}_{\text{tr}}$, Solve equation 4 to get $h_{\alpha,\gamma}, \phi_{\alpha,\gamma}$
12     Compute $\nu := \text{HSIC}_{\gamma}^{\boldsymbol{u}_{\text{valid}}}(\phi_{\alpha,\gamma}(\mathbf{X}), \widehat{\mathbf{V}}^p)$
13     **foreach** $j \in 1 \cdots 100$ **do**
14        Create a permutation $\Phi_j := \phi_{\alpha,\gamma}(\mathbf{X}_{\{\pi(i)\}})$
15        Compute $\nu_j := \text{HSIC}_{\gamma}^{\boldsymbol{u}_{\text{valid}}}(\Phi_j, \widehat{\mathbf{V}}^p)$
16     **end**
17     Compute Thresh $= 99.5^{th}$ percentile of $\{\nu_j\}_{j \in 1 \ldots 100}$
18     **if** $\nu \leq$ *Thresh* **then**
19        ValidHyperparams $:= [\text{ValidHyperparams}, (\alpha, \gamma)]$
20     **end**
21 **end**
22 **foreach** $\alpha, \gamma \in$ *ValidHyperparams* **do**
23     Compute the $\text{AUROC}_{\alpha,\gamma}$ on $\mathcal{D}_{\text{valid}}$
24 **end**
25 Pick $f^* = h^*(\phi^*)$ with the highest $\text{AUROC}_{\alpha,\gamma}$
26 **return** $f^*$

---

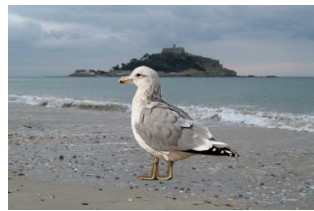 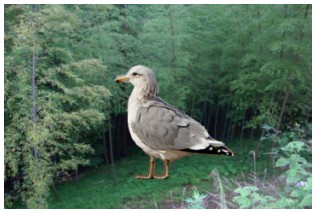 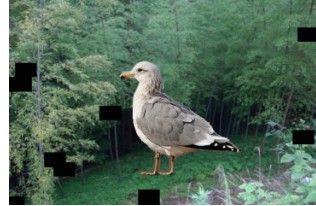

Figure 3: Examples of the generated waterbirds images. Left: water bird on water background. Middle: water bird on land background. Right: water bird on land background with camera artifacts.

---

**Procedure 1:** IdentifySufficientSet

**Input** : $\mathcal{D}, \boldsymbol{\lambda}$
**Output** : $\widehat{V}$

/* Conduct test #1 in proposition 2 */

1   $\boldsymbol{V}^{\underline{d}} := \emptyset$

2   **foreach** *auxiliary label* $V^i \in \mathbf{V}^d$ **do**

3     $p\_val_i = \text{KCIT}(Y, V^i, \texttt{conditioningVariables} = \boldsymbol{V}^{d \backslash i})$

4     **if** $p\_val_i < 0.001$ **then**

5      $\boldsymbol{V}^{\underline{d}} = [\boldsymbol{V}^{\underline{d}}, V^i]$

6     **end**

7   **end**

/* learn the low dimensional representation, $s^* \to \boldsymbol{V}^{\underline{d}}$ */

8   Split $\mathcal{D}$ into $\mathcal{D}_0, \mathcal{D}_1$

9   $s^* = \text{Cross-validation}(\mathcal{D}_0, \boldsymbol{\lambda})$

10   Compute $S := \{s^*(\mathbf{x}_i)\}_{i \in \mathcal{D}_1}$

/* Conduct test #2 in proposition 2 */

11   $\widehat{V} := \emptyset$

12   **foreach** *auxiliary label* $V^i \in \boldsymbol{V}^{\underline{d}}$ **do**

13     $p\_val_i = \text{KCIT}\big(S, V^i, \texttt{conditioningVariables} = (\boldsymbol{V}^{\underline{d} \backslash i}, Y)\big)$

14     **if** $p\_val_i < 0.001$ **then**

15      $\widehat{V} = [\widehat{V}, V^i]$

16     **end**

17   **end**

18   **return** $\widehat{V}$

---

# D   Waterbirds experiments details

## D.1   Data generation

We use the subset of the places images provided by Makar et al. [30] in `https://github.com/mymakar/causally_motivated_shortcut_removal`. We generate the data as follows. $V^{p0} \sim$ Binomial$(0.5)$, $V^{p1}$ is generated such that it has a 70% correlation with $V^{p0}$. $\mathbf{V}^c$ is drawn from Binomial$(0.01)$. We generate the outcome $Y = \sigma(\theta_0 + \theta_{p0}V^{p0} + \theta_{p1}V^{p1} + \boldsymbol{\theta}_c^\top \mathbf{V}^c + \varepsilon)$, where $\sigma(.)$ is the sigmoid function, and $\varepsilon \sim \mathcal{N}(0, 0.5)$.

For $P_s$: $\theta_0 = -0.84, \theta_{p0} = 0.84, \theta_{p1} = 0.4$ and $\boldsymbol{\theta}_c \sim \mathcal{N}(0, 1)$. For $P_{\text{Flip}}$: $\theta_0 = 0.45, \theta_{p0} = -0.84, \theta_{p1} = -0.4$ and $\boldsymbol{\theta}_c \sim \mathcal{N}(0, 1)$. For $P^\circ$: $\theta_0 = -0.15, \theta_{p0} = 0, \theta_{p1} = 0$ and $\boldsymbol{\theta}_c \sim \mathcal{N}(0, 1)$.

Examples of the generated images are in figure 3.

## D.2   Training details

We split the data into 70% training and validation and 30% is a held out test set. The training and validation data is further split into 75% training and 25% validation. We resize the images to a

resolution of $128 \times 128$, and train for 500 epochs. For all HSIC based models, we cross validate over bandwidth values $= [1.0, 10.0, 100.0, 1000.0]$, and $\alpha$ values $= [1e3, 1e5, 1e7, 1e9]$. We picked this set of bandwidths to cross validate over using the following heuristic: for each HSIC model, we train its corresponding unpenalized (i.e., $\alpha = 0$) model. We evaluate the HSIC of the unpenalized model at various bandwidth levels, and pick the set that has non-zero HSIC as the reasonable set to cross validate over.

For all $L2$ models we cross validate over $L2$ penalty $= [0, 0.0001]$. We use Adam optimizer, with the default learning rate $0.001$ and default $\epsilon = 1e - 07$.

Each model takes roughly 50 minutes to train, with of 56 models per simulation and a total of 560 models, the total compute time is roughly 470 hours on a Tesla T4 GPU.

## E    Diabetic Retinopathy details

Similar to waterbirds, we split the data into 70% training and validation and 30% is a held out test set. The training and validation data is further split into 75% training and 25% validation. We follow [21] in preprocessing the images such that they are macula-centered, and resize them to be $299 \times 299$.

We train each model for 2 epochs (which is sufficient since the DR data is larger than the waterbirds data).

We use Adam as our optimizer, and follow tensorflow guidance in setting $\epsilon = 0.1$. We also find that a slower learning rate leads to better results for all models, so set it to $0.0001$.

For the HSIC based model, we consider bandwidths $= [0.1, 1.0]$, which were picked using the same heuristic described in the waterbirds experiment, and $\alpha = [1e3, 1e5, 1e7]$.

Each model takes roughly 30 minutes. With 10 models per simulation, we have 100 models which take a total of 5 hours to train on a Tesla T4 GPU.