# OpenReview forum: "Causally motivated multi-shortcut identification and removal"
_NeurIPS.cc/2022/Conference — NeurIPS 2022 Accept_

### Official Review · Reviewer_1iAR · 2022-07-11

**Rating:** 5
**Confidence:** 3
**Soundness:** 2 fair
**Presentation:** 2 fair
**Contribution:** 2 fair

**Summary:**

This paper aims to deal with the shortcut learning to alleviate distribution shift problem by leveraging the causal structure of a problem. The paper first defines several models and concepts including the casual DAGs, generalization risk, P-specific shortcuts, etc. Then, it identifies a sufficient subset of shortcuts with the help of two properties stated in section 4. Afterward, it builds risk invariant predictors on the given identified set. Experiments on two datasets demonstrate the effectiveness of their method.


**Questions:**

(1) In equation(1), it seems the target set of distributions is calculated from a specific DAG. Whether the arguments and results below relies on a correct DAG?
(2) In section 6.1, it says "our approach is able to correctly identify the two true auxiliary labels which mark the true shortcuts in all 10 simulations". Is there any detailed result?

**Limitations:**

Yes

**Strengths And Weaknesses:**

strength:
(1) The paper provides an idea that robustness to a large set of distribution shifts can be achieved by ensuring the invariance to a small set of shortcuts.
(2) The paper further develops a method to identify the small set of shortcuts.

weakness:
(1) The author proposed a two-step method to build risk invariant predictors, however there lacks statements on the impact of the accuracy of step 1 on step 2. Such as, in section 3, the author assume that Vp is a subset of Vd. How will it affects the robustness of the predictors when it is violated? Since this method is based on a group of complex assumptions, I believe that such results is needed(no matter theoretical or experimental).

(2) The experimental results are indirect to demonstrate the contribution claimed by the authors. The authors claim that their methods are able to identify shortcuts, while in both experiments there lacks an actual result to illustrate it, and merely overall AUROCs on different settings are offered.
(3) It seems that both experiments are synthetic, so it is a little not convincing.

---

> ### Author Response · Authors · 2022-08-01
> **Errors in identifying $V^p$ and empirical evaluation**
>
> We thank the reviewer for their time and thoughtful feedback.
>
> **1. Theoretical or empirical estimate of the impact of having errors in $\hat{V}^p$:**
>
> (a) _Empirical results on sensitivity to identifying $V^p$_: two of our baselines highlight what happens if $\hat{V}^p$ is a poor estimate of $V^p$: W-HSIC-FullV shows what happens if $\hat{V}^p$ includes redundant auxiliary labels, while W-HSIC-HDX shows what happens if $\hat{V}^p$ excludes important shortcuts. While these baselines are outperformed by our main approach, they still outperform all others despite relying on a partially flawed estimate of $V^p$.
>
> (b)  _Theoretical results_: As we state on lines 216-217, our approach for estimating $V^p$ is asymptotically consistent, and hence should converge to the true $V^p$. In addition, results from [6] can be applied to our setting to show that the generalization error bound of our proposed estimator has a fourth order (i.e., mild) dependence on the error in estimating $\hat{V}^p$. We will add this statement to the main paper.
>
> **2. Indirect empirical evaluation:**
> We stress that our main goal is estimating a robust predictor. We view identifying the shortcuts as a means to an end, rather than a goal in and of itself. For this reason, we believe that reporting the AUROCs across distribution shifts is our main result. However, we also report our approach’s ability to identify the correct sufficient shortcuts in lines 340-347. Specifically, we report that our approach identifies the correct sufficient shortcuts in all 10 simulations, while an approach that skips our s-reduction step, identifies the correct shortcut in 1/10 of the simulations.
>
> **Both experiments are synthetic** : First, we note that our experiment setup falls in line with existing literature on distribution shifts. Our waterbirds experiment is done on a commonly accepted and widely used robustness benchmark dataset (e.g., see [1-5] among others). Our second experiment on the diabetic retinopathy data reflects a slightly more realistic medical example. Second, we wish to highlight the difficulty with using non-semi-synthetic data. Evaluating the robustness of our approach requires testing it on multiple test sets with varying $P(V^p, Y)$. In real datasets, this can be achieved by undersampling subpopulations in each of the test sets, which leads to varying sample sizes between different test sets. If we observe that a model’s accuracy has high variance on one test set compared to another, it could be due to poor model performance, or it could be due to one test set being smaller than the other, making it hard to draw meaningful conclusions about the model’s distributional robustness.
>
> **Target distribution is consistent with DAG:**
> The reviewer is correct in that we require the observed data to satisfy our assumptions about the DAG in section 3. This means that some domain knowledge is needed to ensure that the data generating process conforms to our assumptions. However, we believe that such knowledge is readily available for a large number of research questions, e.g., in prediction problems where the task is classifying objects in an image or detecting the presence of a disease from medical imaging. Future work investigating sensitivity to violations in our assumptions is an exciting new direction for research.
>
> **Additional details regarding our statement "our approach is able to correctly identify the two true auxiliary labels which mark the true shortcuts in all 10 simulations"**:
> As we explain in lines 305-306, we simulated the data such that the only 2 true shortcuts are image background and camera artifacts. In addition to those 2 shortcuts, we simulate 10 redundant auxiliary labels that do not mark true shortcuts since they do not affect X. Our approach is able to identify the image background and camera artifacts as the two relevant shortcuts in all 10 simulations.
>
> [1] Sagawa et al. Distributionally robust neural networks for group shifts: On the importance of regularization for worst-case generalization. ICLR 2020
>
> [2] Liu  et al. Just train twice: Improving group robustness without training group information. International Conference on Machine Learning. ICML, 2021
>
> [3] Creager, E., et al. Environment inference for invariant learning. ICML 2021
>
> [4] Sohoni, N., et al No subclass left behind: Fine-grained robustness in coarse-grained classification problems. Neurips 2020.
>
> [5] Makar et al. Causally motivated shortcut removal using auxiliary labels. AIStats 2022.
>
> [6] Foster, Dylan J., and Vasilis Syrgkanis. "Orthogonal statistical learning." 2019

---

### Official Review · Reviewer_3Y9R · 2022-07-15

**Rating:** 6
**Confidence:** 3
**Soundness:** 3 good
**Presentation:** 4 excellent
**Contribution:** 3 good

**Summary:**

The paper proposes a method for (1) idetifying shortcuts among a set of candidate shortcuts and (2) training a predictor that is invariant to these shortcuts. The method is based on assumptions about the causal relationships between the target variables $Y$, observed inputs $X$ and the candidate spurious attributes $V^{d}$, and comes with theoretical guarantees. The authors perform experiments on two semi-synthetic vision datasets.

**Questions:**

**Question 1.** In line 104, could you please clarify what you mean by $P_s(V^6) P_s(Y) \ll P_s(V^p, Y)$? Does the $\ll$ here mean "absolutely continuous with respect to" or  "much lower"? In the latter case, one probability measure cannot be much lower than another one everywhere, as they have to both integrate to $1$?

**Question 2.** In proposition 1 you say that you require $Y$ to be independent from $V^p$ but in the interpretation in line 140 you say $X$ independent from $V^p$. Is this a typo?

**Question 3.** In definition 1, what is $f$? Should it be $g'$?

**Question 3.** You mention "overlap" several times in the paper, e.g. in line 103 and line 148. What exactly do you mean by overlap?

**Question 4.** Could you please comment on the similarity of your work to iCaRL [1]?

**Question 5.** Once the shortcuts are identified, do I understand correctly that you could apply some of the standard methods in the spurious correlation literature to learn a robust predictor? For example, Group DRO?

**Question 6.** How did you select the strength of $L_2$ regularization for the $L2$ method? It appears that it performs poorly even in-distribution on Waterbirds.

[1] *Invariant Causal Representation Learning for Out-of-Distribution Generalization*;
Chaochao Lu, Yuhuai Wu, José Miguel Hernández-Lobato, Bernhard Schölkopf

**Limitations:**

No issues.

**Strengths And Weaknesses:**

I want to note that I am not an expert on causal inference, so I did not verify all the details of the mathematical derivations.

**Strength 1.** The paper is well-written and the presentation is clear. In particular, Section 3 is written in a way that it can be understood by non-expert readers, which I appreciate: the authors provide examples associated with different types of causal structures.

**Strength 2.** The proposed method tackles a challenging setting where the list of shortcuts is not known a priori, and has to be identified from a broader range of candidate shortcuts.

**Strength 3.** The experiments show that all the components of the proposed method are actually useful, and the method cannot be trivially simplified.

**Strength 4.** The authors describe the details of the proposed procedure carefully, including how the cross-validation is performed.

**Weakness 1.** The proposed method is quite complicated and involves a lot of moving parts. As far as I understand, you need to (1) train a predictor $s(x)$ that predicts the attributes $V^d$ from $X$; (2) train a predictor $\eta(v^p_i, y_i)$ in order to obtain example weights to achieve independence between $Y$ and $V^p$; (3) train a feature representation $\phi(X)$ and classifier $h(x)$; (4) perform multiple independence tests during shortcut identification, feature representation learning and validation. Given the complexity of the procedure, it is likely that it may not work out of the box on a new problem.

**Weakness 2.** Experiments only compare the variations of the proposed method, standard training with weight decay and group weighting. I understand that the paper tackles a unique setting, but it would be good to include other baselines such as Group DRO (given that shortcuts are known) or methods such as Just Train Twice, even if the comparison is not entirely fair. Group DRO could possibly provide an upper bound on the robust classifier performance with identified shortcuts.

**Weakness 3.** Experiments are all done on semi-synthetic problems. Both problems use artificially added black patches as one of the shortcuts. It would be nice to see a more realistic application of the proposed method.

**Typos**:
- line 7: interprability
- line 92: are are
- line 166: two tests that enables
- line 170: in principal
- line 281: the that

---

> ### Author Response · Authors · 2022-08-01
> **Baselines and real data experiments**
>
> Thank you for your thoughtful review!
>
> **Weakness 1, Complexity of the method**: We will add pseudo code for our algorithm in the appendix, and we will make our code publicly available to help with implementation.
>
> **Weakness 2, Group DRO and Just train twice (JTT) as a baseline:** As we explain in lines 338-339, Group DRO is very closely related to (and in some cases equivalent to) baselines that we have implemented (W-L2-FullV, W-L2-S and W-L2-HDX). All these baselines are outperformed by our approach.
> JTT does not leverage the auxiliary labels. In settings like ours where they are available, not leveraging them is a missed opportunity. The JTT paper itself confirms that, showing that models with access to $V^d$ (specifically, DRO) outperform JTT. Since our approach outperforms methods roughly equivalent to DRO, we have good reason to believe that our approach will outperform JTT.
>
> **Weakness 3, Semi-synthetic experiments:** First, we note that our experiment setup falls in line with existing literature on distribution shifts. Our waterbirds experiment is done on a commonly accepted and widely used robustness benchmark dataset (see [1-5] among others). Our second experiment on the diabetic retinopathy data reflects a slightly more realistic medical example. Second, we wish to highlight the difficulty with using non-semi-synthetic data. Evaluating the robustness of our approach requires testing it on multiple test sets with varying $P(V^p, Y)$. In real datasets, this can be achieved by undersampling subpopulations in each of the test sets, leading to varying sample sizes between different test sets. If we observe that a model’s accuracy has high variance on one test set compared to another, it could be due to poor model performance, or it could be due to one test set being smaller than the other, making it hard to draw meaningful conclusions about the model’s distributional robustness.
>
> **Question 1 and 3b, $P_s(V^p)P_s(Y) << P_s(V^p, Y)$ and overlap**: This notation means absolutely continuous with respect to, we will clarify that. As we state on lines 103-104, it reflects the overlap assumption, which is common in the causal literature. Intuitively, it can be understood as an assumption that $P_s(Y=y, V^p=v^p)$ is non-zero for all $y, v^p$. We will explicitly state that.
>
> **Question 2, $Y$ or $X$ independent to $V^p$**: We thank the reviewer for catching this typo. Line 140 should read $Y$ (not $X$) independent to $V^p$, which is consistent with proposition 1.
>
> **Question 3a**: We thank the reviewer for catching the typo in definition 1, $f$ should be $g$.
>
> **Question 4, similarity to iCARL**: We thank the reviewer for pointing out this related work, we will refer to it in the related works section. The work in iCARL is fundamentally different from ours in that they assume access to multiple datasets collected from different environments. We assume access to a single dataset collected from one environment. In addition, since one of their main goals is identifying causal factors controlling $Y$, they impose assumptions on the functional form of some components of the probability distribution (that they belong to an exponential family). We do not impose any such assumptions, and allow the true distribution to take any form.
>
> **Question 5:** _“Once the shortcuts are identified, could you apply standard methods in the spurious correlation literature…[like] group DRO?_
>
> In principle, yes. However,  the algorithm that the reviewer suggests (identifying the shortcuts then deploying groupDRO) is equivalent to our baseline W-L2-S, and is outperformed by our suggested approach. This is because our suggested approach has an additional HSIC penalty, which reduces the variance of the estimator without introducing bias (as we show in the appendix, proposition A1).
>
> **Question 6:** The strength of the L2 regularization is picked via cross-validation as outlined in the appendix, section C lines 545-546. The fact that our approach does better in distribution is consistent with findings from Makar et al 2022, who show that the MMD penalty leads to improved efficiency compared to the typical L2 penalty even in-distribution. The same conclusion holds for our HSIC penalty, as we highlight in the appendix, proposition A1.
>
> We will also correct the other typos that the reviewer identified.
>
> [1] Sagawa et al. Distributionally robust neural networks for group shifts: On the importance of regularization for worst-case generalization. ICLR 2020
>
> [2] Liu  et al. Just train twice: Improving group robustness without training group information. International Conference on Machine Learning. ICML, 2021.
>
> [3] Creager, E., et al. Environment inference for invariant learning. ICML 2021
>
> [4] Sohoni, N., et al No subclass left behind: Fine-grained robustness in coarse-grained classification problems. Neurips 2020.
>
> [5] Makar et al. Causally motivated shortcut removal using auxiliary labels. AIStats 2022.

---

> > ### Comment · Reviewer_3Y9R · 2022-08-08
> > **Thank you for your response**
> >
> > Thank you for your response and clarifications! I am satisfied with the response and maintain my score.
> >
> > Could you please clarify the connection between the baselines you implemented and Group DRO? I think it would be useful to include the camera ready version of the paper as well. Generally, I would recommend including a more detailed description of the baselines in the appendix, as these are not standard methods that the readers will be familiar with.

---

> > > ### Author Response · Authors · 2022-08-09
> > > **Group DRO**
> > >
> > > Thank you for your comment.
> > > Group DRO is equivalent to our method when the loss is convex, but this equivalent might break down when they are non-convex.
> > > This is explored in detail in the group DRO paper [1] on page 8 under theoretical comparison.
> > >
> > > [1] Sagawa, "Distributionally Robust Neural Networks for Group Shifts: On the Importance of Regularization for Worst-Case Generalization" ICLR 2020, https://arxiv.org/pdf/1911.08731.pdf

---

### Official Review · Reviewer_4zx1 · 2022-07-16

**Rating:** 7
**Confidence:** 4
**Soundness:** 4 excellent
**Presentation:** 4 excellent
**Contribution:** 3 good

**Summary:**

The paper uses auxiliary labels to detect and mitigate shortcuts. The key idea is that they are able to go from a large set of auxiliary labels to a potentially small set that are actually shortcut labels. The most of the representations building follows from Makar et al. 2022.






**Questions:**

1. Selecting the auxiliary labels seems to be as hard as finding spurious features using expert knowledge. How can I pick auxiliary variables without knowledge?
2. While V^d is allowed to be highdimensional, and then try to find a subset by testing each V separately. Conditional independence testing is supposed to be hard when things get high-dimensional (queue the advent of CRTs https://arxiv.org/abs/2104.10618 etc.). Can the authors comment on if and why they believe this may not be an issue? Is KCIT particularly suited here?

related issue:

Finding s(X) in equation 2 seems to require building a model with a high-dimensional output (if V^d is allowed to be high-dimensional). Was this easy in practice? How large can V^d get before this step seems a large error?

3. Can the authors address the main concern from above? Giving an example would be greeat here.

**Limitations:**

See questions and weaknesses.

**Strengths And Weaknesses:**

The paper is easy to read and makes an important contribution to an important problem. They take clean and useful steps to go from having auxiliary labels that are not necessarily shortcut labels to get to a shortcut labels by using conditional independence tests. This seems to suggest that we can pick up a bunch of labels about the task and use them as V^d. The experiments in the paper are encouraging. Lines 105-113 talking about a specific important problem were useful and informative!


My main issue with this paper is the ability to collect auxiliary labels that satisfy the conditions of $V_c$ in figure 1. My issue is that one can end up collecting auxiliary variables that are causal children of $X^*$. This would mean that s(X) is dependent on these always. This means these potentially invariant and informative features (because they appear in $X^*$) would end up being listed in $\hat{V}_p$. Then, under  means going to $P_0$ will make Y independent on some invariantly relevant features. This would lead to suboptimal error.

This means that a lot of care could still be required in specifying $V_c$, which would require domain knowledge or some knowledge of shortcuts. That said, I do believe the paper points out a useful next step to do when one is able to get a list $V_d$ that does not specify any $X^*$ related features.

I have other issues with the claim of the ability to handle high-dimensional $V_d$. See questions.


There's a few papers that have related causal/anticausal factorizations and learning that could maybe go in the related work because they also handle high-dimensional shortcuts or do not require exact knowledge of the shortcut features: causal sufficiency based representation learning https://arxiv.org/abs/2109.03795, equation 1 seems to like the family here and high-dimensional spurious features  https://arxiv.org/abs/2107.00520, the causal diagram is used along with environments but not spurious features https://arxiv.org/abs/2006.07500, use augmentations to build robust models https://arxiv.org/abs/2111.12525


===== Update after rebuttal

The authors have addressed my concerns. I'm updating my score accordingly.

---

> ### Author Response · Authors · 2022-08-01
> **Violations to assumptions and high dimensional auxiliary labels**
>
> We thank the reviewer for their constructive feedback.
>
> **1- Domain knowledge:** _“My issue...is the ability to collect auxiliary labels that satisfy the conditions on $V^c$”, “a lot of care could still be required in specifying $V^c$”,_ and  _“[other papers] do not require exact knowledge of the auxiliary labels”._
>
> We stress that we do not require exact knowledge of the shortcut labels. We only require access to a set of auxiliary labels ($V^d$) that satisfy the assumptions in section 3, but do not require the user to distinguish between meaningful shortcuts ($V^p$), and redundant features ($V^c$). Proposition 2 establishes that we can distinguish the two subsets without domain knowledge, by conducting 2 sets of independence tests.
>
> **2- Violations to the assumption about auxiliary labels** : _“One can end up collecting collecting auxiliary labels that are children of $X^*$”_, and _“selecting auxiliary labels seems to be as hard as finding spurious features using expert knowledge”_
>
> Like any other ML algorithm, our approach might have suboptimal performance under violations of our assumptions. However, we believe that the domain knowledge required to ensure our assumptions are not violated is readily available for a large number of important research questions. E.g.,  in prediction problems where the task is classifying objects in an image or detecting the presence of a disease from medical imaging. In this case, almost any auxiliary data that is associated with the image cannot be a causal descendant of the image itself: for example X-ray pixels ($X$) are unlikely to be the causal parent of the auxiliary variables $V^d$ since X-ray pixels cannot cause a person to have pneumonia, be a woman or be a certain age. The case where $X^*$ is a causal parent is even less likely. Intuitively, $X^*$ is a sufficient statistic (i.e., a summary) of the causal effect of $Y$ on the appearance of the image $X$, for example opacity in the appearance of the lungs in case of detecting pneumonia. Changes in the appearance of an image caused by the target label are unlikely to be the causal parent of an auxiliary label. Overall, we agree with the reviewer that our work represents important first steps, and that future work should focus on developing sensitivity analysis for settings where our assumptions are violated. We will state that in the paper.
>
> **3- Conditional independence testing vs. KCIT**: _“Conditional independence tests are supposed to be hard when things get high dimensional”_.
>
> The reviewer is correct. This is exactly why we propose conducting the independence tests on $s(X)$ which is a lower dimensional projection of $X$. We chose KCIT for ease of implementation since it is relatively well known among the ML community, and because of its strong theoretical guarantees (asymptotically consistent). Other independence tests might be suitable, and would be interesting to explore in future work, we will state that.
>
> **4 - If $V^d$ is high dimensional, will $s(x)$ be high dimensional?**: Not necessarily. As outlined in line 191-197, we first remove the subset $V^d$ that satisfies the condition $Y \perp_{P_s} V^i \mid V^{i\/d}$. For the remaining variable set, which we denote by $\underline{d}$, we train $s(x) \rightarrow V^\underline{d}$. While $V^\underline{d}$ likely has a smaller dimension than $V^d$, it might still be high dimensional, making the output from $s(x)$ high dimensional. However, we note that there is a vast literature on high dimensional classification which shows that the task of learning $s(x)$ has a mild dependency (up to logarithmic) on the size of $\underline{d}$ [1], which means that $s(x)$ can be learned efficiently from finite (small) samples. We did not find practical challenges with this step.
> Note: we will fix the typo in the definition of s(x) on line 196, it should read $s(x) : X → V^\underline{d}$ not $s(x) : X → V^d$.
>
> **5- Related work**: We thank the reviewer for pointing out additional related work, we will include these papers in the related works section.
>
> [1] Lei, Yunwen, et al. "Generalization error bounds for extreme multi-class classification." CoRR, abs/1706.09814 (2017).

---

> > ### Comment · Reviewer_4zx1 · 2022-08-07
> > **Thanks for the response**
> >
> > I have updated my score.

---

### Author Response · Authors · 2022-08-01
**Summary of reviews**

We thank the reviewers for their thoughtful feedback. We are encouraged that they recognize the importance of our contribution (Reviewers 4zx1 and 3Y9R), the clarity of our presentation (Reviewers 4zx1 and 3Y9R) and the strength of our experimental findings (Reviewers 4zx1 and 3Y9R).

Two of the reviewers asked about the performance of our approach in real data settings.
First, we note that our experiment setup falls in line with existing literature on distribution shifts. Our waterbirds experiment is done on a commonly accepted and widely used robustness benchmark dataset (e.g., see [1-5] among others). Our second experiment on the diabetic retinopathy data reflects a slightly more realistic medical example. Second, we wish to highlight the difficulty with using non-semi-synthetic data. Evaluating the robustness of our approach requires testing it on multiple test sets with varying $P(V^p, Y)$. In real datasets, this can be achieved by undersampling subpopulations in each of the test sets, which leads to varying sample sizes between different test sets. If we observe that a model’s accuracy has high variance on one test set compared to another, it could be due to poor model performance, or it could be due to one test set being smaller than the other, making it hard to draw meaningful conclusions about the model’s distributional robustness.

In addition, two of the reviewers asked about how our method relates to other work. We thank the reviewers for pointing us to relevant literature. We recognize that with such a fast growing topic, it is difficult to include a comprehensive literature review. However, we believe including the suggested citations will make our paper stronger.

We respond to additional questions by each reviewer separately below.


[1] Sagawa et al. Distributionally robust neural networks for group shifts: On the importance of regularization for worst-case generalization. ICLR 2020

[2] Liu  et al. Just train twice: Improving group robustness without training group information. International Conference on Machine Learning. ICML, 2021

[3] Creager, E., et al. Environment inference for invariant learning. ICML 2021

[4] Sohoni, N., et al No subclass left behind: Fine-grained robustness in coarse-grained classification problems. Neurips 2020

[5] Makar et al. Causally motivated shortcut removal using auxiliary labels. AIStats 2022

---

### Meta-Review · Area_Chair_Ex9b · 2022-08-20

**Recommendation:** Accept
**Confidence:** Certain

**Metareview:**

The reviewers agreed the paper is a worthwhile contribution in a growing area of identifying and removing shortcuts for robustness to distributional shift. Please take the reviewers feedback into consideration for the camera-ready.

**Award:**

No

---

### Decision · Program_Chairs · 2022-09-14

Accept